# Mathematical modeling and performance evaluation of Ducted Horizontal-axis Helical Wind Turbines: Insights into aerodynamics and efficiency

Zishan Shaikh[1], Ahmad Fazlizan[1]*, Halim Razali[1], Kok Hoe Wong[2], Altaf Hossain Molla[3], Rabiu Aliyu Abdulkadir[4,5], Dumitru Baleanu[6,7]*, Rabha W. Ibrahim[6,8,9]*

1 Solar Energy Research Institute, Universiti Kebangsaan Malaysia, Bangi, Selangor, Malaysia, 2 Carbon Neutrality Research Group (CNRG), University of Southampton Malaysia, Iskandar Puteri, Malaysia, 3 Department of Mechanical and Manufacturing Engineering, Faculty of Engineering and Built Environment, Universiti Kebangsaan Malaysia, Bangi, Selangor, Malaysia, 4 Faculty of Information Science and Technology, Universiti Kebangsaan Malaysia, Bangi, Selangor, Malaysia, 5 Department of Electrical Engineering, Aliko Dangote University of Science and Technology, Wudil, Nigeria, 6 Department of Computer Science and Mathematics, Lebanese American University, Beirut, Lebanon, 7 Institute of Space Sciences, Magurele-Bucharest, Romania, 8 Information and Communication Technology Research Group, Scientific Research Center, Al-Ayen University, Thi-Qar, Iraq, 9 Department of Mathematics, Mathematics Research Center, Near East University, Nicosia, Mersin 10-Turkey

* a.fazlizan@ukm.edu.my (AF); dumitru.baleanu@gmail.com (DB); rabhaibrahim@yahoo.com (RWI)

## Abstract

With the escalating demand for energy, there is a growing focus on decentralized, small-scale energy infrastructure. The success of new turbines in this context is notable. However, many of these turbines do not follow many of the basic ideas established to evaluate their performance, leaving no precise technique or mathematical model. This research developed a Ducted Horizontal-axis Helical Wind Turbine (DHAHWT). The DHAHWT is a duct-mounted helical savonius turbine with a venturi and diffuser to improve flow. Unlike a vertical axis helical savonius turbine, DHAHWT revolves roughly parallel to the wind, making it a horizontal turbine. This complicates mathematical and theoretical analysis. This study created a DHAHWT mathematical model. COMSOL simulations utilizing Menter's Shear Stress Transport model (SST) across an incoming velocity range of 1m/s to 4m/s were used to evaluate the turbine's interaction with the wind. MATLAB was used to train an artificial neural network (ANN) utilizing COMSOL data to obtain greater velocity data. The Mean Average Percentage Error (MAPE) and Root Mean Square Error (RMSE) of ANN data were found to be 3%, indicating high accuracy. Further, using advanced statistical methods the Pearson's correlation coefficient was calculated resulting in a better understanding of the relationship of between incoming velocity and velocity at different sections of the wind turbine. This study will shed light on the aerodynamics and working of DHAHWT.

**Data Availability Statement:** All relevant data are within the manuscript and its Supporting Information files.

**Funding:** The author(s) received no specific funding for this work.

**Competing interests:** The authors have declared that no competing interests exists.

## 1. Introduction

One of the great difficulties of the twenty-first century is ensuring reliable, economical, and environmentally sustainable energy supplies. Energy infrastructure is at the heart of this dilemma, serving as a point of convergence for a wide range of policy objectives ranging from economic growth and national security to climate change mitigation and social inequity [1]. As the energy demand increases in the world a surge in the demand for small and more efficient energy systems is seen, notably within the realm of wind energy [2]. Many researchers contributed to this pursuit and introduced systems that can help in achieving high efficiency. Some have worked on adding a duct in order to increase the mass flow of air through the turbine, like initial development done by Sabzevaria in 1977 [3] and further development to the latest wind lens by Yuji Ohya and Takashi Karasudani in 2010 [4], Chong et al. Omnidirectional Guided Vanes (ODGV) [5] and to the latest INVELOX system, studied by by Allaei et al. in 2014 [6] and further developed by Wanlong Han et al. in 2015 [7]. Further some researchers have taken another interesting way, including wind energy recovery from man-made sources of wind [5, 8–14], introducing novel and biomimetic blades [15, 16], Smart blades [17], or bladeless turbines designed by corporations like Magenn Power (Lighter than air) and Vortex Bladeless Turbine (Skybrator). All these efforts have aided in the increase in efficiency with a significant reduction in size and noise.

Within the landscape of technological advancements, the maturation of wind energy technology into energy recovery technology emerges as a noteworthy development, underscored by its multifaceted advantages. Chief among these merits is the copious availability of wind energy sources, affording a sustainable and abundant reservoir for power generation. Moreover, the technology capitalizes on the consistent and elevated wind speeds, thereby augmenting its operational efficiency and dependability. This confluence of propitious factors establishes wind energy as a promising and ecologically sound solution for addressing the escalating demands of our energy landscape.

Apart from the above-mentioned benefits, these systems have also changed the way different empirical and semi-empirical models would fit the analysis of the system. In 2007, Van Bussel [18] had developed a theoretical model based on Betz's theory establishing a direct relationship between power augmentation and mass flow increase. Hansen [19] proved that the power coefficient increase is proportional to the increase in the ratio of the mass flow through the shrouded turbine to that through the bare turbine. Apart from the above-mentioned theories there are many theories that are used in the evaluation of the performance of a wind turbine. These models are categorized into two types: semi-empirical models for the near wake and semi-empirical models for the far wake. Their categorization is based on their effectiveness in assessing the aerodynamic efficiency of blades and forecasting the flow pattern in the distant wake zone. Semi-empirical models take into account the forces exerted on the rotor, aiding in power output prediction. Additionally, they consider blade shape, which directly influences both aerodynamic blade performance and the wake near the rotor region.

A simple model in the under semi-empirical near wake models is the Blade element momentum which suggests the calculation of the load on the turbine by dividing the turbines into small sections [20]. An extension of the same is the actuator disc theory, which suggests that a blade is a semi-permeable disk at the rotor plane and the downwash velocity from the turbine reduces roughly by 66% when compared to the free stream velocity. The actuator disk theory when coupled with Computational Fluid Dynamics (CFD) can help calculate the flow-field in the wake [21]. However, these theories face significant challenges in accurately calculating the output for certain novel turbines featuring a long central axis, a velocity gradient within

the turbine, and multiple types of wind velocities, highlighting the anomalous nature of their design that poses practical difficulties for precise predictions.

Certainly, even Ducted Wind Turbines, which are well-suited for small-scale applications in urban settings, face a lack of understanding of their operational principles. This results in a shortage of reliable and efficient methodologies for analysis and design. [22].

This paper aims at deriving a novel method for the calculation of power output for DHAHWT which would consider the different forms of velocities within the turbine range as well as the velocity gradient throughout the turbine. This is necessary as it would help in understanding the not just the overall performance but also the aerodynamics related change in efficiency. The proposition made in this research paper is based on the data collected from CFD simulation in stationary state, which was further used in Artificial Neural Network (ANN) to generate more data and finally evaluated for Pearsons's coefficient.

## 2. Theoretical overview of wind turbine performance

The status quo for the mechanical power generation from a wind turbine is based on the Betz theory (1926). This model was developed based on linear momentum theory. Linear Momentum theory is based on the analysis of loading on the blades by applying conservation principles of linear and angular momentum. For any turbine performance analysis, it is necessary that a control volume is assumed, in which the boundaries resemble the shape of a stream tube. The status quo is suggested to be a general model that applicable for various wind turbines. The assumptions used were as follows:

- Control volume of the fluid

- Homogenous, incompressible steady state fluid flow

- No frictional drag

- An infinite number of blades

- Uniform thrust over the disk or rotor areas.

- Continuity of velocity through the disk

- A non-rotating wake

- The static pressure far upstream and far downstream is equal to the undisturbed ambient static pressure.

Utilizing the principle of preserving linear momentum within the control volume encompassing the entire system allows for the determination of the overall force acting on the system. In adherence to the law of mass conservation, the quantity of air passing through various segments of the stream tube remains consistent [23]. Since kinematic models, often referred to as explicit models, are straightforward wake models that can be solved analytically, they are appropriate for assessing the wind speed deficiencies in sizable wind farms. These models, such as those by Jensen, Larsen (first order and second order), and Frandsen, simulate the velocity deficit in the wake behind a turbine using momentum equations [24, 25]. Notably, these models do not consider the initial expanding zone of the wake. Fig 1 illustrates the development of the wake. These models rely on the initial velocity profile of the near-wake region to calculate the velocity deficit and wake expansion using the simplified momentum equations based on these velocity profiles. To consider, the change in the intensity of turbulence these models must be coupled with a turbulence model as the original forms of the models cannot evaluate such advance modelling. The Larsen model, one of the previously mentioned models,

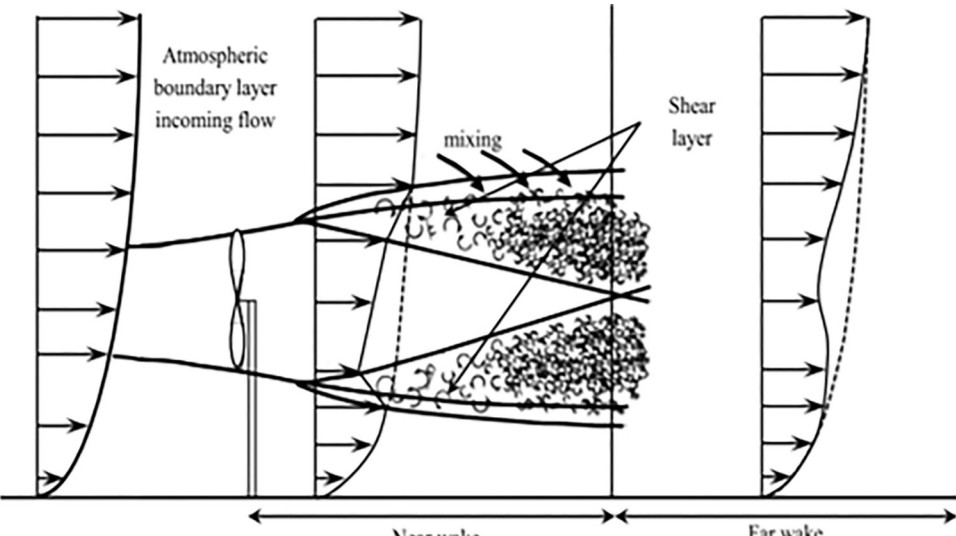

**Fig 1. Schematic representation of wind turbine wake when turbine is placed in an atmospheric boundary layer** [27].

can consider the impact of ambient turbulence on wake expansion. The governing equations needed for each of these kinematic far-wake models form the basis of the simulations. In comparison to a boundary layer that is highly turbulent, a laminar ABL has longer far-wake areas and a higher velocity deficit. Lower energy is available to the wind turbines further downstream because of the longer wake. However, due to the fluctuating stresses that come from increased turbulence, this may also increase the rate of turbine blade wear. On the other side, increased turbulence intensity speeds up wake recovery [26].

The development of Diffuser-Augmented Wind Turbines (DAWTs) has expanded over 50 years without commercial success. Initially, the conversion process was poorly understood, and experimentalists focused on maximizing pressure drop over the rotor disk without theoretical guidance. Increasing the diffuser area and negative back pressure at the exit proved beneficial. Claims of performance enhancements by a factor of 4 or more were unconfirmed. A simple momentum theory showed that power augmentation in DAWTs is proportional to mass flow increase at the nozzle, achievable by adjusting the diffuser exit ratio and reducing negative back pressure. The theory predicts an optimal pressure drop of 8/9, the same as for bare wind turbines, with maximum energy extraction per unit of volume. Practical experiments have not achieved power augmentation factors above 3, although theoretical rotor power coefficients of CP, rotor = 2.5 may be achievable at the cost of larger diffuser area ratios (typically with β > 4.5).

## 3. Methodology

### 3.1 Energy balance for DAWT

The change in energy for an open system of flow of fluid through a duct with isothermal conditions and no work done is given by:

$$\Delta E = \frac{P_1}{\rho} + \frac{V_1^2}{2} - \frac{P_2}{\rho} + \frac{V_3^2}{2} \tag{i}$$

To calculate the change in energy when work is done can be calculated by combining Eq i and wind energy equation.

$$\Delta E = \frac{P_1}{\rho} + \frac{V_1^2}{2} - \left[ \left( \frac{P_2}{\rho} + \frac{V_3^2}{2} \right) + \frac{1}{2} \rho A V_1^3 \right] \tag{ii}$$

Considering the hypothesis of Actuator disk theory,

$$\Delta E = \frac{P_1}{\rho} + \frac{V_1^2}{2} - \left[ \left( \frac{P_2}{\rho} + \frac{(V_1 - 0.66 \langle V_1 \rangle)^2}{2} \right) + \frac{1}{2} \rho A V_1^3 \right] \tag{iii}$$

Considering the hypothesis of One-dimensional momentum theory, we get.

$$\Delta E = \frac{P_1}{\rho} + \frac{V_1^2}{2} - \left[ \left( \frac{P_2}{\rho} + \frac{(\gamma(1-a)V_0)^2}{2} \right) + \frac{1}{2} \rho A V_1^3 \right] \tag{iv}$$

According to actuator disk theory, the velocity of the wind flow drops by $\frac{2}{3}$ (66.66%) after passing through the wind turbine. Further, as per the one-dimensional momentum theory in a DAWT, this drop may be less as the optimum axial induction factor being $\frac{1}{3}$. However, regardless of the extent of the velocity drop, it would significantly reduce the exit velocity and cause a negative impact on the Energy Recovery System causing major failure and other issues.

Fig 2 presents a comprehensive diagram detailing the aerodynamic analysis of turbines using Computational Fluid Dynamics (CFD) simulation, specifically with COMSOL software. The systematic approach includes CFD simulation, data collection, and machine learning correlation analysis, notably employing Pearson correlation.

## 3.2 DHAHWT wind turbine

The DHAHWT denotes a ducted helical wind turbine characterized by wind flow alignment along its central axis. This configuration classifies the turbine as a horizontal axis wind turbine due to the alignment with the wind direction. Despite previous employment as a vertical axis wind turbine in open natural wind conditions, this marks its inaugural utilization within a duct in a horizontal axis orientation.

The innovative implementation of this system aims to exploit hitherto neglected wind energy resources generated by anthropogenic sources. Conventional wind turbine designs are unsuitable for this purpose, as their application could compromise system efficiency. Thus, the necessity for an inventive turbine design with distinct aerodynamic properties becomes evident.

In instances involving a novel wind turbine design that deviates from prevalent assumptions, the applicability of currently employed equations may exhibit limitations in accurately capturing the turbine's operational outcomes. Subsequent sections will delve into the intricate wind behavior in the vicinity of the turbine. A comprehensive undrstanding of this phenomenon mandated the execution of a steady-state CFD investigation for the DHAHWT.

To facilitate the simulation, the Reynolds Averaged Navier-Stokes Equation (RANS) was employed, utilizing Menter's Shear Stress Model (SST). The choice of RANS stemmed from the study's objective to grasp the cumulative effects arising from wind-turbine interaction, transcending mere eddy current observations. Similarly, the selection of SST was predicated on its amalgamation of k-ε and k-ω turbulence models, yielding enhanced visualization of both wall-bounded and free-stream flow characteristics.

For a given simulation, it is necessary to have an appropriate $y^+$ value which is a dimensionless parameter. This dimensionless value helps to classify the types of boundary layers. $y^+$ is

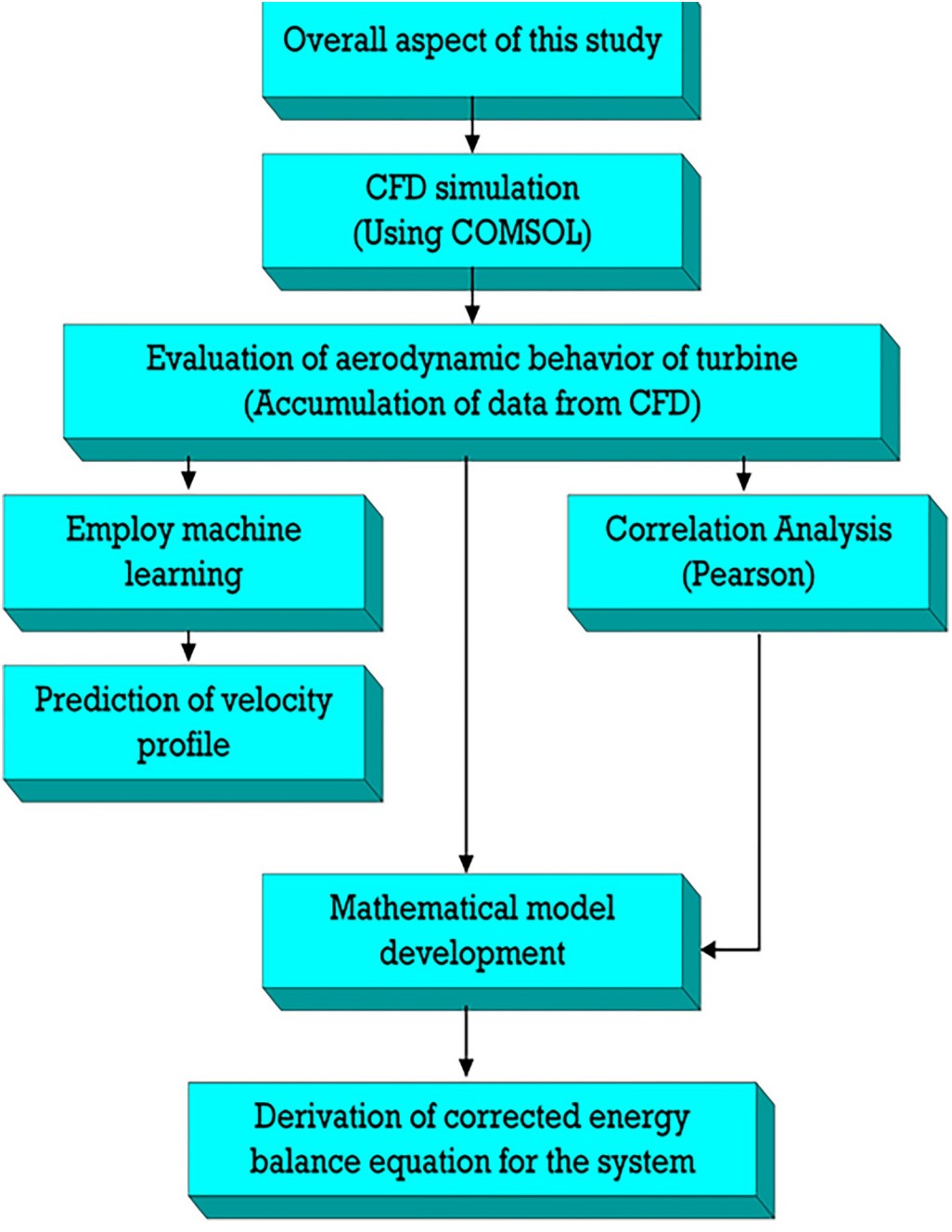

**Fig 2. Flow of study and methodology.**

given by

$$y^+ = \frac{u.y}{v}$$

Where, u is the shear velocity at the nearest wall, y is the absolute cell distance from the nearest wall and $v$ is the kinematic viscosity. A value of $y^+$ less than 5 is considered as sub laminar. However, a $y^+$ value of more than 5 is excellent for 3D visualization of simulation.

### 3.3 Simulation analysis

A Boundary condition is the interaction between the system and the environment. The boundary conditions in a fluid dynamics study include velocity inlet and outlet, wall (no-slip/slip) etc. For the topic of study following were the boundary conditions.

- The domain of the turbine was selected to be solid.

- The domain of the duct was selected to be non-solid.

- The fluid material selected was Air at equilibrium discharge.

- Properties of the fluid were user-defined to 30∘ Celsius.

- The material at the duct walls was selected as Polyvinyl Chloride (PVC) material with roughness selected.

- At the wall, NO SLIP condition was applied.

- Inlet and outlet conditions were specified.

- Outlet pressure was set adjusted to align in accordance with the ambient pressure to account for the back pressure.

- Inlet velocity was for range 1 m/s to 4 m/s.

The simulation was carried out considering the flow to be incompressible range hence the continuity equation would apply [28].

$$\rho(\mathbf{u} \cdot \nabla)\mathbf{u} = \nabla \cdot [-\rho\mathbf{I} + \mathbf{K}] + \mathbf{F} \tag{v}$$

$$\rho\nabla.\boldsymbol{u} = 0 \tag{vi}$$

Here, 'u' is the fluid velocity, p is the fluid pressure and ρ is the fluid density. 'F' is the external forces that are fluid. However, in equation 1 no additional forces have been added hence F can be neglected. The value of K in equation 1 is given by:

$$\mathbf{K} = (\mu + \mu_{\mathrm{T}})(\nabla\mathbf{u} + (\nabla\mathbf{u})^{\mathrm{T}}) \tag{vii}$$

$$\rho(\mathbf{u} \cdot \nabla)\mathrm{k} = \nabla \cdot [(\mu + \mu_{\mathrm{T}}\sigma_k)\nabla\mathrm{k}] + P - \beta_0^*\rho\omega k \tag{viii}$$

$$\rho(\boldsymbol{u} \cdot \nabla)\omega = \nabla \cdot [(\mu + \mu_T\sigma_\omega)\nabla\omega] + \frac{\gamma}{\mu_T}\rho P - \rho\beta_0\omega^2 + 2(1 - f_{v1})\frac{\sigma_{\omega2}\rho}{\omega}\nabla k \cdot \nabla\omega, \ \omega = om \tag{ix}$$

Here, 'om' is the specific dissipation rate and 'k' is the turbulent kinetic energy.

$$\nabla G \cdot \nabla G + \sigma_w G(\nabla \cdot \nabla G) = (1 + 2\sigma_w)G^4, \ \ell_\omega = \frac{1}{G} \cdot \frac{\ell_{ref}}{2} \tag{x}$$

Here, 'G' is the reciprocal wall distance.

$$\mu_T = \rho\frac{a_1 k}{max(a_1\omega, Sf_{v2})}, S = \sqrt{2\boldsymbol{S} : \boldsymbol{S}}, \boldsymbol{S} = \frac{1}{2}\left(\nabla\boldsymbol{u} + (\nabla\boldsymbol{u})^T\right) \tag{xi}$$

Fig 3 shows the diagram of the setup drawn in COMSOL and meshed. For the meshing of the setup a physics-controlled mesh was generated. This was the case as it can generate the best meshing with the negligible error in the meshing. Further a mesh independent study was carried out to rule of the influence of the size of the mesh on the results generated. Here, the

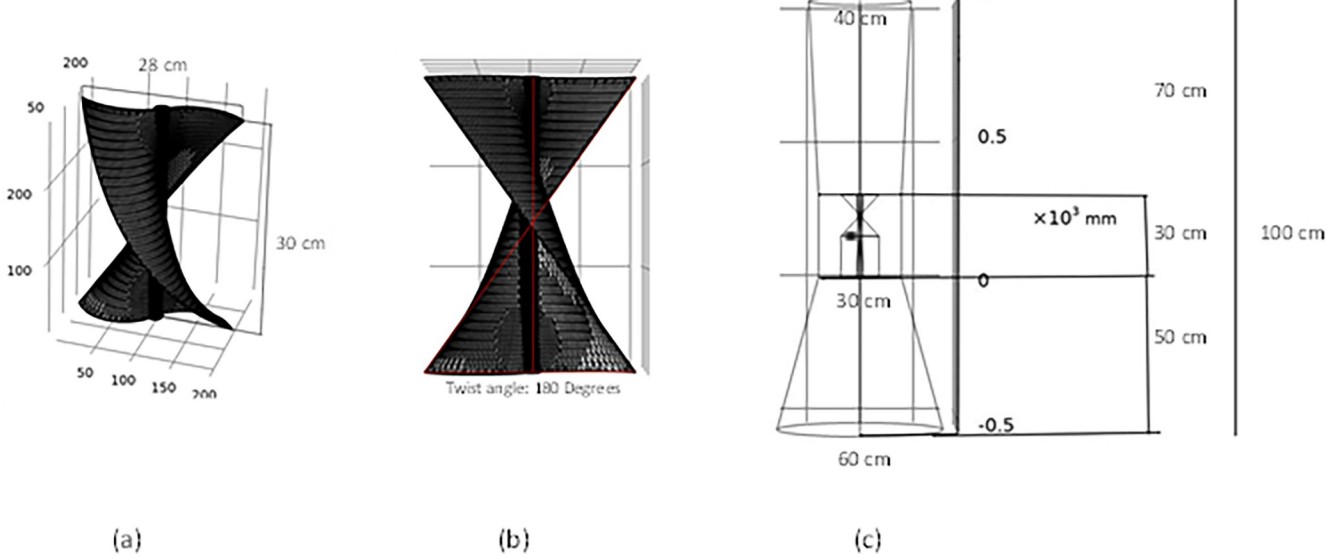

**Fig 3.** (a) The meshing and geometry of the helical turbine (b) Meshing and twist angle of the turbine in COMSOL (c) Geometry of the complete system.

minimum element size is 0.9, maximum element size is 13.8 with the maximum element growth being 1.08. The overall mesh quality was 0.234. A mesh independent study was conducted to rule out the possibility of results being dependent on the mesh size. Further a grid convergence study suggests that the error is lower than $10^{-1}$, which is an industrially acceptable result. This can be seen in Fig 4.

### 3.4 Data collection and processing

Artificial Neural Networks (ANNs) have demonstrated versatility in wind energy research in 2020, Acarer et al. explored radial wind turbines' (RWT) adaptability for residential use [29].

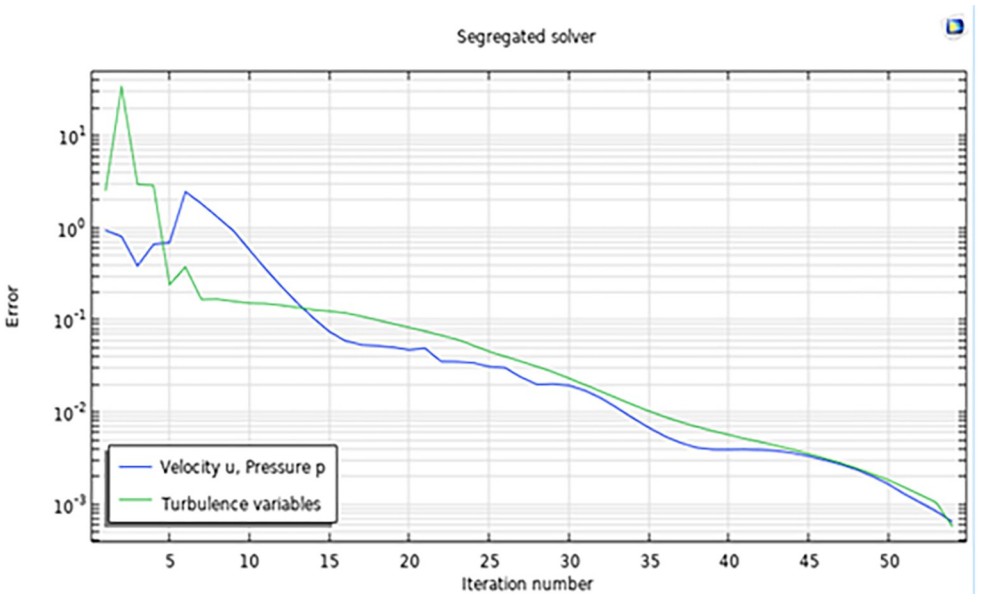

**Fig 4. Error study.**

An exterior casing serves as the entry point for the airflow, which interacts tangentially with the turbine blade before leaving the casing through a chimney. In 2018, Wang et al. used a suggested design with preliminary optimizations utilizing CFD simulations and response surface modelling with applications of machine learning algorithms [30]. Additionally, Sarkic-Glumac et al. conducted meta-modelling with assistance from an artificial neural network (ANN) for shape optimization. Astonishingly, the 3D model's Cp value of 0.29 at = 0.55 indicated a 103% increase in Cp over the designs that had previously recorded the maximum efficiency [31]. In this work, the evaluation of wind profile around the turbine is evaluated at various velocities. For the training set velocity data range from 1m/s to 3.5 m/s was evaluated. This data was obtained from COMSOL simulation. Further, the data of 4m/s velocity was used as a test range and later the data for 6m/s velocity was predicted.

All the basic training data set was imported from the simulations conducted in COMSOL. The data collected was at different points on the turbine (6 points across the diameter and 30 along the axis) at velocities ranging from 1m/s to 3.5m/s. The points can be seen in Fig 5 Further, to evaluate the testing, the data obtained from COMSOL was compared with the prediction data obtained from ANN. The data was further evaluated using MAPE, RMSE and MBE based on the following equations. The evaluation of these statistical methods is done using the

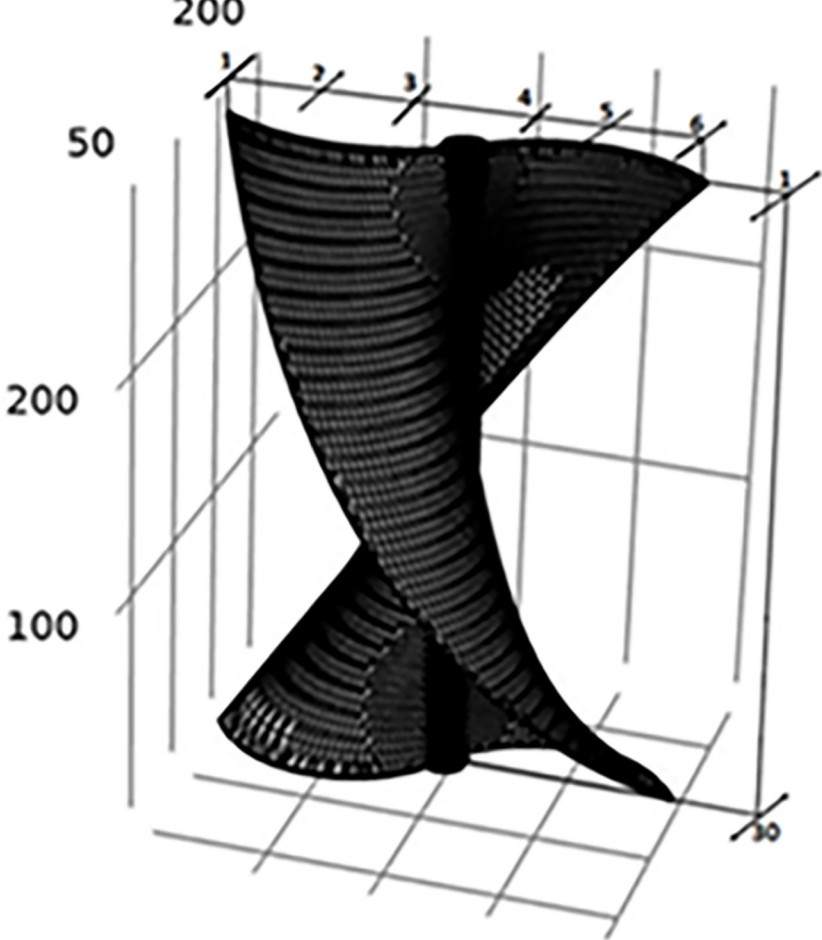

**Fig 5. 6 Points across the diameter and 30 points along the axis for data.**

following equations:

$$MAPE = \frac{1}{n}\sum_{i=1}^{n}\frac{V_{sim} - V_{ANN}}{V_{sim}}$$

$$MAPE = \sqrt{MAPE = \frac{1}{n}\sum_{i=1}^{n}\left(V_{sim} - V_{ANN}\right)^2}$$

$$MBE = \frac{1}{n}\sum_{i=1}^{n}\left(V_{sim} - V_{ANN}\right)^2$$

Here, $V_{sim}$ is the velocity obtained from COMSOL and $V_{ANN}$ is the data predicted data, and n denotes the total tested datapoints [32]. Elevated precision is associated with reduced (MAPE), (RMSE), and (MBE) values. The Artificial Neural Network (ANN) model underwent evaluation using MAPE as a classification metric, with a MAPE of less than 10% indicating a high degree of accuracy. After this classification, the mean MAPE was calculated for each model, enabling the determination of a model suitable for turbines of differing dimensions.

### 3.5 Data processing

In this research investigation, a thorough correlation analysis was conducted utilizing the Python programming language to explore the associations between two pivotal variables: the turbine's inlet velocity and velocities at various designated points and sections within the turbine system, as illustrated in Fig 4. The primary aim of this study is to ascertain the underlying nature of the relationship between the variables. To achieve this objective, multiple datasets obtained from simulation-based studies and Artificial Neural Networks (ANN) were subjected to advanced statistical analysis to compute the correlation coefficient. Specifically, Pearson's correlation coefficients were computed in this context to assess both the strength and direction of the relationships that exist among these variables.

## 4. Results

### 4.1 Simulation

From Fig 6 it is known that the velocity just before the turbine ($V_1$) is equal to the velocity after the turbine ($V_2$). This voids the postulates stated by the Actuator Disk theory as no reduction in the velocity is seen after the turbine. This may be attributed to the behavior of air through the turbine. It can also be seen that throughout the turbine there is a gradient of velocity. This would suggest that the thrust throughout the turbine would be different. The curvature of the velocity suggests that multiple types of velocities are present in the turbine region and it imperative that all these types of velocities be considered.

The velocities to be considered include the axial velocity (Va) and tangential velocity (Vc). The Va and Vc lie within the turbine, where Va is seen near the central axis and Vc between the central axis and the underside of the blade edge. To understand the profile better it was necessary that more data be obtained about the velocity profile. The confirmation of the same can be seen in the next section.

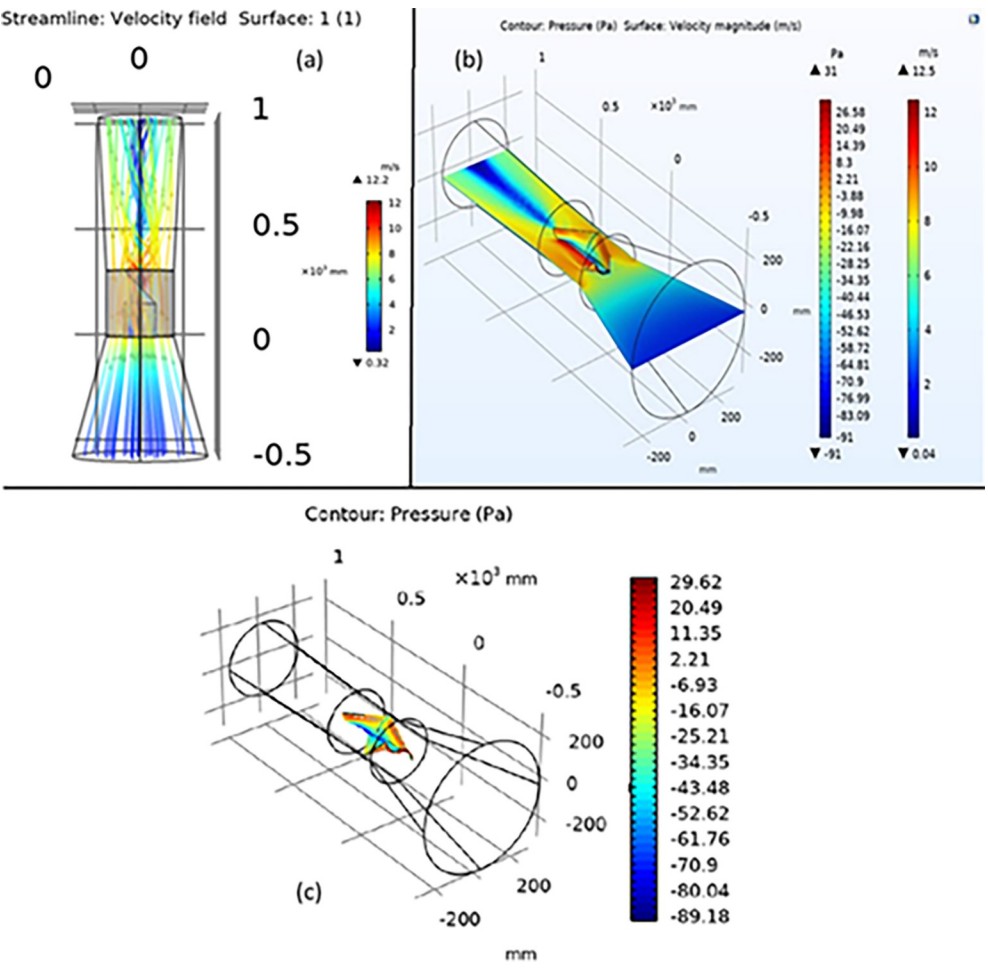

**Fig 6.** (a) Streamline velocity depiction through the ducted Horizontal Axis-Helical Wind Turbine system. (b) Velocity and Pressure relation on a cut plane through the ducted Horizontal Axis-Helical Wind Turbine system. (c) Pressure on the turbine defining the velocity profile across the turbine.

## 4.2 Artificial Neural Network

From Fig 7 it is known that a velocity gradient is present across the diameter and length of the turbine. It can also be noted that the velocity profile at the outermost edge of the turbine is seen dropping. However, the case for the velocity profile near the axis of the turbine is seen decreasing until the first 20% of the turbine followed by a sharp increase. Later the velocity profile is seen stabilizing. It can also be seen that the velocity profile in the mid-section of the turbine shows a very marginal increase in the velocity profile followed by a very small decrease in the velocity to the end of the turbine. The behavior of the wind near the axis is such due to the stagnation point being very close to the axis in the initial section of the turbine and the turbine acting as a funnel beyond the 10% of the turbine.

Using the inputs from the simulation as targets the system was trained nine times to verify the accuracy of data stream. The use of data from the simulation makes it easy to acquire data which was previously impossible to attain and provides information into wind-turbine interaction and aerodynamics. This also proves that similar irregularly shaped turbines can be analyzed for their aerodynamics at a range of velocities based on basic inputs from simulations. It has been seen that the tick and trial method has been effective and has helped in optimizing

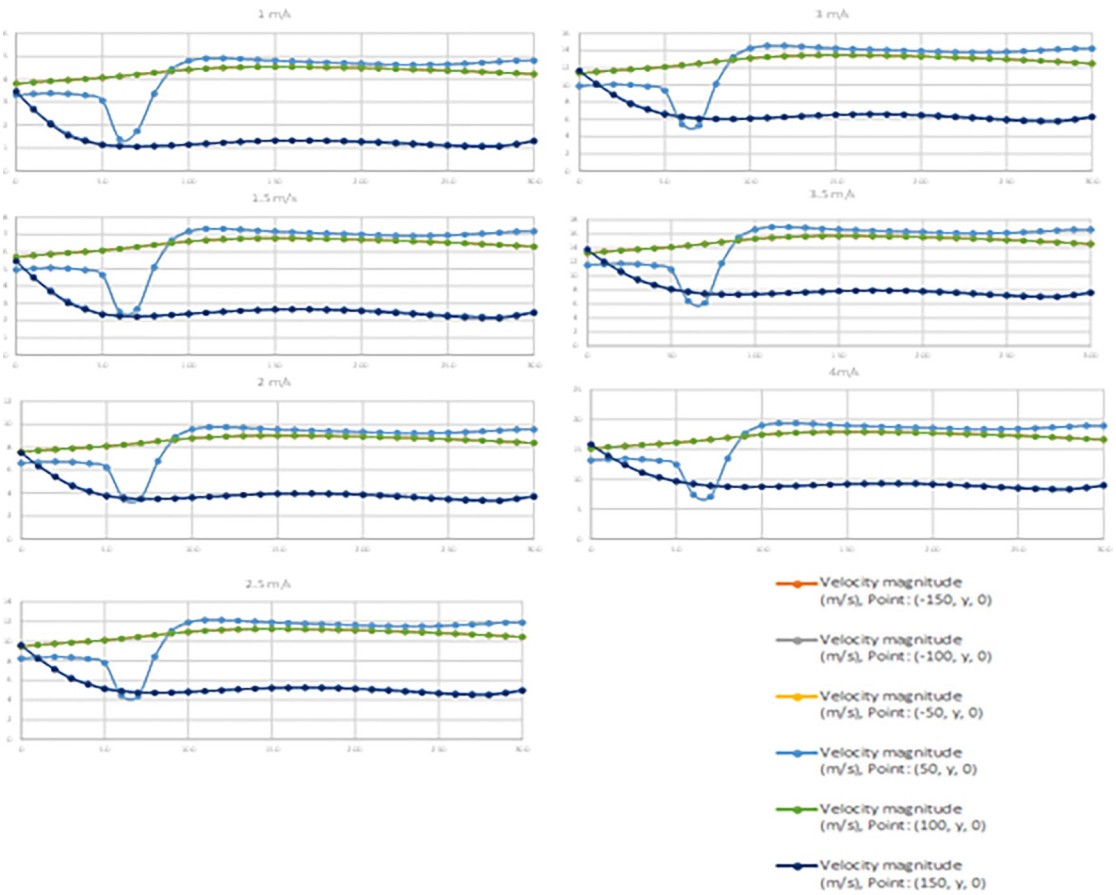

**Fig 7. Velocity gradient throughout the turbine at various points and velocity.**

the neural network. The system was tested and predicted for velocity ranges when the incoming wind velocity was 4 m/s and 6 m/s, as outlined in Table 1, indicating the error in the ANN-generated data.

Further, to understand the difference between the simulation based and the ANN based data the velocity profile at each of the diameter section were plotted. It can be seen from Fig 8 that both the simulated and predicted data The system was tested and predicted for velocity ranges when the incoming wind velocity was 4 m/s and 6 m/s, as outlined in Table 1, indicating the error in the ANN-generated data. This is well in agreement with the MAPE obtained earlier.

**Table 1. Simulation data evaluation metrics.**

| Points | MAPE | MBE | RSME |
|---|---|---|---|
| 150 | 2.41% | 0.18 m/s | 0.32 m/s |
| 100 | 1.53% | 0.01 m/s | 0.30 m/s |
| 50 | 4.02% | 0.86 m/s | 0.45 m/s |
| -50 | 2.78% | 0.51 m/s | 0.17 m/s |
| -100 | 1.77% | 0.37 m/s | 0.21 m/s |
| -150 | 2.36% | 0.26 m/s | 0.05 m/s |

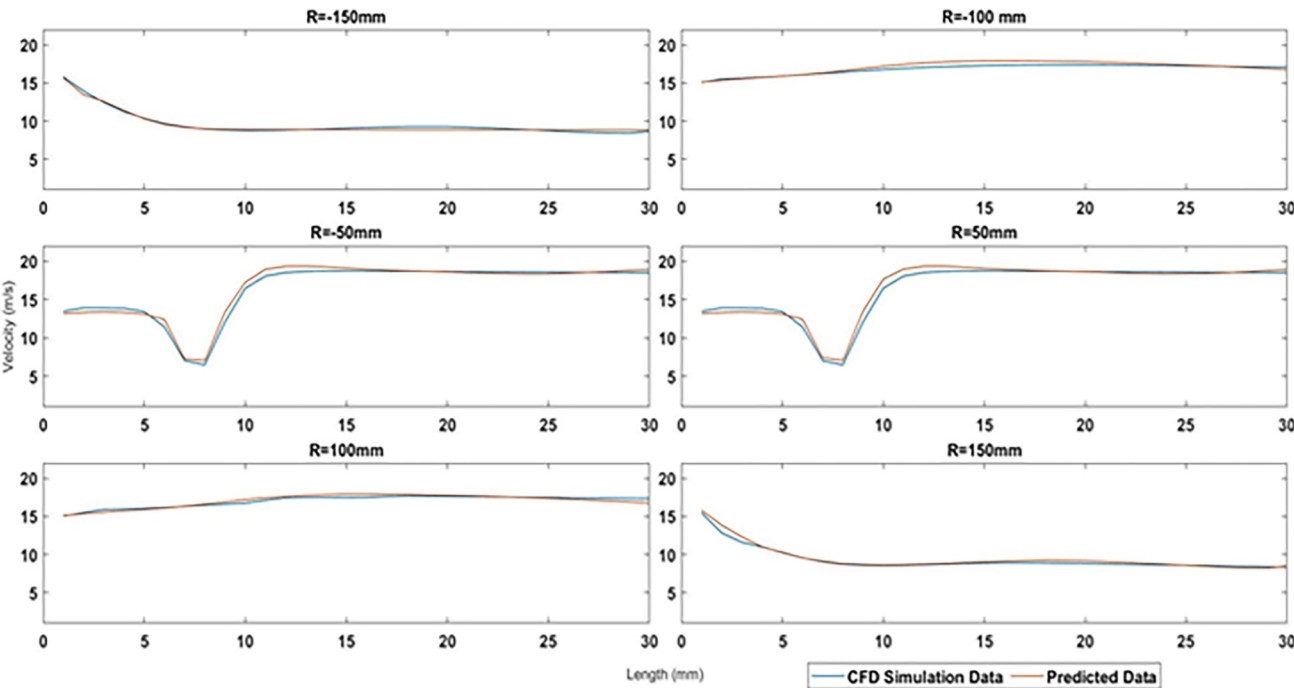

**Fig 8. Simulation based and predicted data co-relation.**

### 4.3 Pearson's correlation

The primary aim of the correlation analysis was to assess and quantify the relationships between the turbine's inlet velocity and the magnitudes of velocity at distinct positions along the diameter of the turbine blade. This analysis utilized a heatmap diagram to visually represent these connections as shown in Fig 9.

Interestingly, the analysis revealed a strong and statistically significant correlation between the inlet velocity and the velocity magnitudes at various positions across the turbine blade's diameter. This finding suggests that changes in the inlet velocity have a direct and measurable impact on the velocity magnitudes at specific locations of the turbine blade. The study also suggests that the velocity is inversely related with the incoming velocity and directly related at certain locations in the turbine. Consequently, modifications in the inlet velocity, whether an increase or decrease, correspondingly lead to alterations in the velocity magnitudes at these positions.

Moreover, the velocity magnitudes at discrete points spanning the diameter of the turbine blade exhibit noticeable interrelationships. This observation indicates that changes in velocity magnitude at one point tend to be mirrored by similar changes at other points along the blade's diameter. A consistent pattern of variation in velocity magnitude prevails across the entirety of the blade's diameter.

### 4.4 New mathematical model

Fig 10 introduces the indexes used at various locations. Here, the ambient pressure is noted as P0, V1 is the velocity just before the turbine and P1 is the pressure just before the turbine, P2 and V2 are the downwash pressure and velocity of the turbine, respectively. Finally, the P3 and V3 are the exit pressure and velocity, respectively. To understand more about the flow, the type of flow must be determined, which in this condition is the incompressible flow. As the

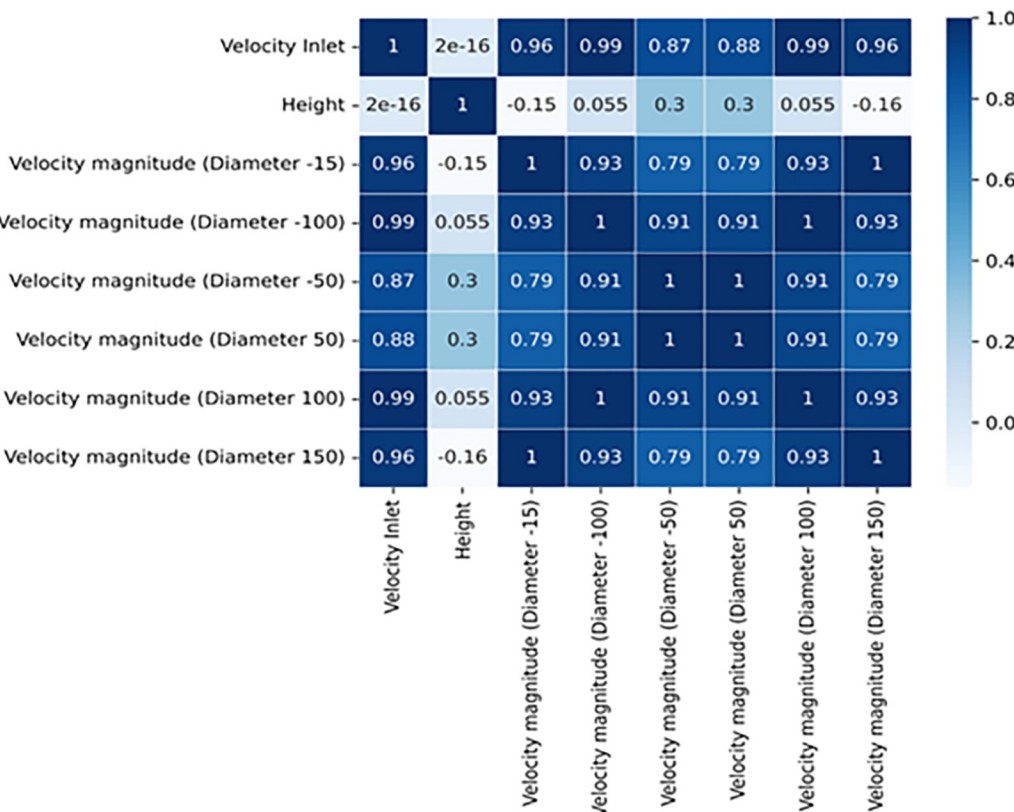

**Fig 9. Statistical evaluation of the relationship between the inlet velocity and velocity at different segments of the turbine.**

flow is incompressible, it is known that the continuity equation can be applied as there is no change in the density. The velocity range can be known from Fig 3, which shows the velocity at different points.

Therefore,

$$P + \frac{1}{2}\rho V^2 = Constant \tag{xii}$$

$$P_1 + \frac{1}{2}\rho V_1^2 = P_2 + \frac{1}{2}\rho V_2^2 = P_3 + \frac{1}{2}\rho V_3^2$$

Fig 11 shows the direction of the wind flow with respect to the turbine and the velocity triangle. As established earlier, it is known that we must consider the velocity triangle in this situation. And hence the equation becomes:

$$P + \frac{1}{2}\rho\left(v_{resultant}\right)^2 = Constant$$

$$Where\ v_{resultant} = \sqrt{v_c + v_a}$$

$v_c$ = tangential velocity
$v_a$ = axial velocity

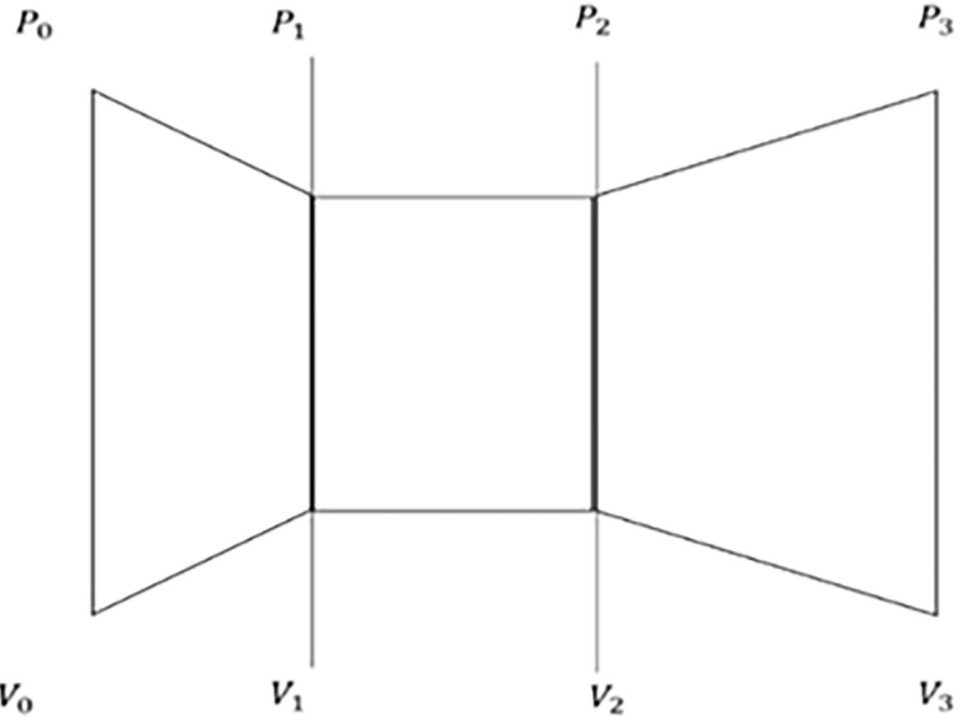

**Fig 10. Pressure and velocity relation in an empty diffuser.**

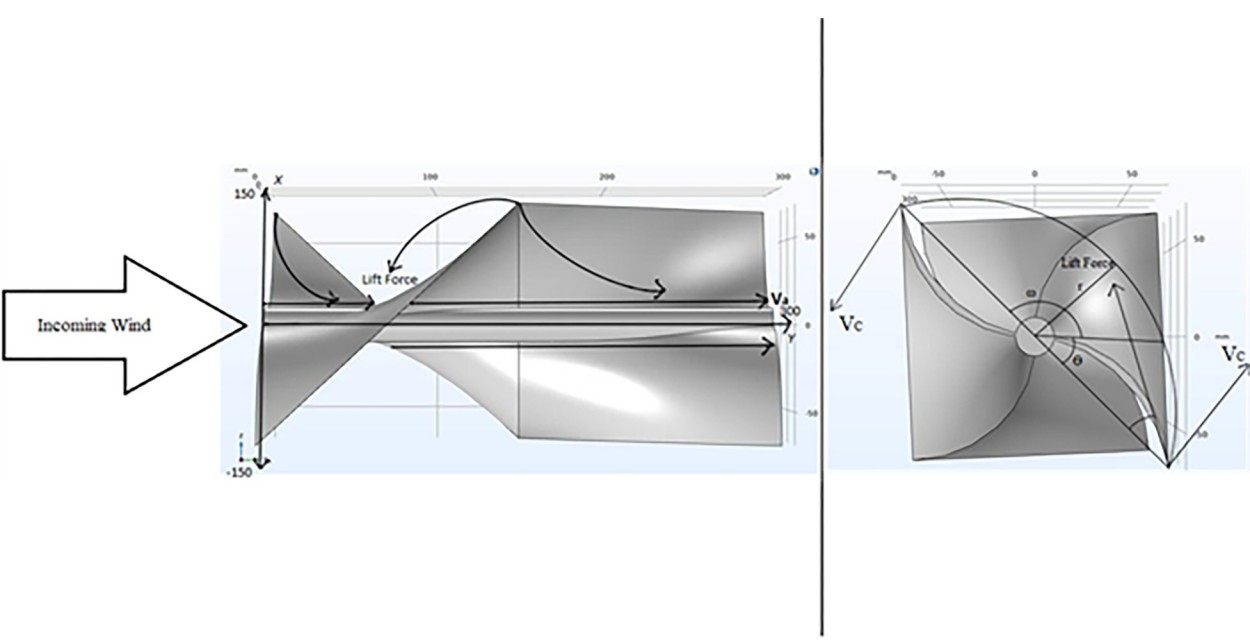

**Fig 11. Blade velocity and force diagram.** Here, ω is the rotation, r is the radius, Θ is the angle.

But,

$$v_c = r\omega$$

Substituting the value of $v_c$ in equation 1, where r = radius (meter) and ω = angular velocity, (rad/sec) which corresponds to the radius of the HHAWT, we get,

$$P + \frac{1}{2}\rho(\sqrt{r\omega + v_a})^2 = Constant \tag{xiii}$$

However, looking at the simulations it is understood that different velocities at different radial points of the turbine. In which case the average of $v_c$ would be considered. Hence,

$$\text{Average } v_c = \frac{v_{c1} + v_{c2} + v_{c3}\ldots\ldots\ldots\ldots v_{cn}}{n}$$

Therefore,

$$Power = \frac{1}{2}A\rho\left(\frac{\sum_i^{30}\left[\sqrt{\left\{\frac{\sum_i^4 v_c}{4}\right\}} + \left\{\frac{\sum_i^2 v_a}{2}\right\}\right]}{30}\right)^3 \tag{xiv}$$

To calculate the pressure profile of the turbine the following equation can be used.

$$P + \frac{1}{2}\rho\left(\frac{\sum_i^{30}\left[\sqrt{\left\{\frac{\sum_i^4 v_c}{4}\right\}} + \left\{\frac{\sum_i^2 v_a}{2}\right\}\right]}{30}\right)^2 = Constant$$

For the calculation of the torque the equation suggested is:

$$Torque = \frac{1}{2}A\rho C_T\left(\frac{\sum_i^{30}\left[\sqrt{\left\{\frac{\sum_i^4 v_c}{4}\right\}} + \left\{\frac{\sum_i^2 v_a}{2}\right\}\right]}{30}\right)^3 \tag{xv}$$

To calculate the lift-force the formula used is:

$$F = C_L Aq\left(\frac{\sum_i^{30}\left[\sqrt{\left\{\frac{\sum_i^4 v_c}{4}\right\}} + \left\{\frac{\sum_i^2 v_a}{2}\right\}\right]}{30}\right)^2 \tag{xvi}$$

Here, $C_L$ is the coefficient of lift, A is the area and q are the dynamic pressure.

Based on the equation and the behavior of the turbine it can be said that the energy balance equation changes to the following:

$$\Delta E = \frac{P_1}{\rho} + \frac{V_1^2}{2} - \left[ \left( \frac{P_2}{\rho} + \frac{(V_1)^2}{2} \right) + \frac{1}{2}\rho A \left( \frac{\sum_i^{30}\left[ \sqrt{\left\{ \frac{\sum_i^4 v_c}{4} \right\}} + \left\{ \frac{\sum_i^2 v_a}{2} \right\} \right]}{30} \right)^3 \right] \qquad \text{(xvii)}$$

From the energy balance equation, it can be noted that the velocity of the wind after the turbine will be the same. This suggests that the Kinetic energy of the velocity is conserved. The development of this energy balance equation suggests that no back force or negative pressure on the system would be seen. This makes the energy recovery system applicable for various exhaust systems.

This culminates in the formulation of an innovative theoretical framework for the Ducted Horizontal-axis Helical Wind Turbine, positing that: "*The theoretical framework posits that the power output of the Ducted Horizontal-axis Helical Wind Turbine system is contingent upon a comprehensive evaluation of diverse wind velocities, encompassing both axial and circular velocities, across all segments of the Ducted Horizontal-axis Helical Wind Turbine*".

## 5. Conclusion

In conclusion, the culmination of this study involves not only the establishment of a novel theoretical framework for the Ducted Horizontal-axis Helical Wind Turbine but also the formulation of a groundbreaking perspective. This theoretical framework posits that the power output of the system is intricately linked to a thorough assessment of various wind velocities, incorporating both axial and circular velocities, across all sections of the Ducted Horizontal-axis Helical Wind Turbine.

Furthermore, this innovative framework, when juxtaposed with the findings related to the Ducted Horizontal-axis Helical Wind Turbine, underscores the need for adaptive theoretical models tailored to the distinct characteristics of different turbine configurations. The traditional Blade Element Momentum (BEM) theory and Actuator Disk Theory, previously deemed standard, are found to be inadequate for the DHAHWT due to its unique features.

The proposed mathematical model, designed for the DHAHWT, not only presents a solution for understanding and optimizing its performance but also holds the potential to drive advancements in turbine design and computational analysis methodologies. As the research progresses, the synergy between these theoretical frameworks and practical applications could redefine our approach to wind turbine engineering, fostering innovation and efficiency in sustainable energy production.

## Supporting information

**S1 Data.**
(XLSX)

## Author Contributions

**Conceptualization:** Zishan Shaikh.

**Data curation:** Zishan Shaikh.

**Formal analysis:** Zishan Shaikh.

**Funding acquisition:** Zishan Shaikh, Dumitru Baleanu, Rabha W. Ibrahim.

**Investigation:** Zishan Shaikh, Ahmad Fazlizan, Altaf Hossain Molla, Rabha W. Ibrahim.

**Methodology:** Zishan Shaikh, Ahmad Fazlizan, Rabiu Aliyu Abdulkadir.

**Project administration:** Ahmad Fazlizan, Halim Razali.

**Software:** Kok Hoe Wong.

**Supervision:** Ahmad Fazlizan, Halim Razali, Kok Hoe Wong.

**Writing – original draft:** Zishan Shaikh.

**Writing – review & editing:** Zishan Shaikh, Ahmad Fazlizan, Altaf Hossain Molla.

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
