## [Decision Letter · Decision Letter 0]

24 Jan 2024

PONE-D-23-38272Mathematical Modeling and Performance Evaluation of Ducted Horizontal-axis Helical Wind Turbines: Insights into Aerodynamics and EfficiencyPLOS ONE

Dear Dr. Shaikh,

Thank you for submitting your manuscript to PLOS ONE. After careful consideration, we feel that it has merit but does not fully meet PLOS ONE’s publication criteria as it currently stands. Therefore, we invite you to submit a revised version of the manuscript that addresses the points raised during the review process.

We look forward to receiving your revised manuscript.

Kind regards,

Ashfaque Ahmed Chowdhury, Ph.D., FHEA, FIEB

Academic Editor

PLOS ONE

Journal Requirements:

3. We note that your Data Availability Statement is currently as follows: 

All relevant data are within the manuscript and its Supporting Information files.

**Comments to the Author**

1. Is the manuscript technically sound, and do the data support the conclusions?

Reviewer #1: Yes

Reviewer #2: No

2. Has the statistical analysis been performed appropriately and rigorously? 

Reviewer #1: Yes

Reviewer #2: N/A

3. Have the authors made all data underlying the findings in their manuscript fully available?

Reviewer #1: Yes

Reviewer #2: No

4. Is the manuscript presented in an intelligible fashion and written in standard English?

Reviewer #1: No

Reviewer #2: No

5. Review Comments to the Author

Reviewer #1: Please take into consideration the attached comments in the PDF file and the following suggestions/questions to further enhance the quality of your work:

- Could you provide insights into the computational resources required for these simulations? How scalable is the model to handle larger or more complex turbine designs?

- I recommend considering a more descriptive heading for "2. Theory Review," perhaps something like "Theoretical Overview of Wind Turbine Performance" to better reflect the content and purpose of this section.

- Consider relocating Figure 2 to the "3. Methodology" section to improve the flow of information.

- The methodology relies on a control volume assumption. How sensitive is the model to variations in the shape and size of the control volume? It wasn't clear to me if different control volume configurations have been explored, and if so, what impact do they have on turbine performance analysis?

- In section 3.3, the fluid pressure is denoted as "p" in the text, but it doesn't appear in any equations. Consider using the "rho" Greek letter to denote fluid density for clarity and consistency in notation.

- How sensitive is the model to variations in temperature, and was the impact of temperature changes on fluid properties considered in the analysis?

- How does the simulation model account for dynamic changes in wind conditions? Are there considerations for transient simulations to capture the turbine's response to varying wind speeds?

- The duct material is specified as Polyvinyl Chloride (PVC). Could you discuss how the choice of duct material influences the simulation results? Were material properties, such as roughness, taken into account in the analysis?

- Some figures in the text appear to be written in lowercase.

- Some sections of the text use the first person ("we"). Consider revising these instances to the third person for a more formal and objective tone in scientific writing

Reviewer #2: The manuscript presents a numerical analysis on a ducted helical with horizontal axis.

- The motivation of the use of this kind of machine is not well described. What is the advantage in comparison with a classical ducted wind turbine? What is the advantage in comparison with a helical wind turbine with a vertical axis?

- The performance of the machine are not shown, the CP-lambda curves are not evaluated.

- The simple mechanical balance on ducted wind turbines are impenetrable, and a significant original contribution lacks.

- The bibliography review on ducted wind turbines misses important and recent works revealing the real potential of this kind of devices.

- The use of an ANN is not motivated.

6. PLOS authors have the option to publish the peer review history of their article (what does this mean?). If published, this will include your full peer review and any attached files.

Reviewer #1: **Yes: **Paulo Mendes

Reviewer #2: No

---

## [Author Response · Author response to Decision Letter 0]

18 Mar 2024

The authors of the paper are thankful to the reviewers. The comments, suggestions and reviews have been understood and implemented in the paper. We hope these suggestions would aid in further improve the quality of the paper and help make an iota of difference in the field.

Reviewer 1

Review Response

Could you provide insights into the computational resources required for these simulations? How scalable is the model to handle larger or more complex turbine designs? 1. The device used for the initial simulation was Dell Latitude 13 7320, 11th Generation i7-Processor, 16GB DDR4 RAM. Later for further curation of data and results the system used was Intel Xenon CPU-E5-26900, 32GB DDR3 RAM, with 8 cores, Logical Processor 16.

2. As of now due to the limitation of computational power of the systems available scalability of the system was not tested. 

3. However, in the future the authors want to test the scalability and the application of the same for the same. 

4. When it comes to complex turbine designs, the authors have evidence to believe that as long as the aerodynamic interactions and behaviour of the complex turbine is same the simulation as well as the mathematical model shall be applicable.

I recommend considering a more descriptive heading for “2. Theory Review” perhaps something like “Theoretical Overview of Wind Turbine Performance” to better reflect the content and purpose of this section. We thank you for the suggestion and accept and implemented the suggestion.

Consider relocating Figure 2 to the “3. Methodology” section to improve the flow of information. We thank you for the suggestion and accept and implemented the suggestion.

The methodology relies on a control volume assumption. How sensitive is the model to variations in the shape and size of the control volume? It wasn’t clear to me if different control volume configurations have been explored, and if so, what impact do they have on turbine performance analysis. The authors have tested the system at different control volumes and the trends of the flow were noted and described in figure 7. However, due to constraints in the computational power the authors could not evaluate the performance based on shape of the control volume. For the same the authors have some experimental data.

In section 3.3, the fluid pressure is denoted as “p”; in the text, but it doesn’t appear in any equations. Consider using the “rho” Greek letter to denote fluid density for clarity and consistency in notation. We thank you for the suggestion and accept and implemented the suggestion.

How sensitive is the model to variations in temperature, and was the impact of temperature changes on fluid properties considered in the analysis? The temperature parameter has not been considered by the authors as temperature would not affect the properties of the fluid unless the temperature fluctuation would be in a range of ± 10○C. The impact of temp is negligible.

How does the simulation model account for dynamic changes in wind conditions? Are there considerations for transient simulations to capture the turbine’s response to varying wind speeds? The current system designed has a unidirectional flow from the inlet at the bottom to the outlet at the top, owing to this the dynamic changes in the wind direction were not taken into consideration. However, an effort to capture the response of the turbine turbulence created by a fan at the inlet was made but the computational requirements were very high.

The duct material is specified as Polyvinyl Chloride (PVC). Could you discuss how the choice of duct material influences the simulation results? Were material properties, such as roughness, taken into account in the analysis? 1. After the computational study a prototype was developed in order to study the experimental performance of the system. The material used for the developed prototype was PVC, hence, making it the choice.

2. Yes, the material property of roughness was taken into account during the simulation.

Some figures in the text appear to be written in lowercase. We thank you for the suggestion and accept and implemented the suggestion.

Some sections of the text use the first person (“we”). Consider revising these instances to the third person for a more formal and objective tone in scientific writing. We thank you for the suggestion and accept and implemented the suggestion.

Reviewer 2

The authors of the paper are thankful to the reviewers. The comments, suggestions and reviews have been understood and implemented in the paper. We hope these suggestions would aid in further improve the quality of the paper and help make an iota of difference in the field.

Review Response

The motivation of the use of this kind of machine is not well described. What is the advantage in comparison with a classical ducted wind turbine? What is the advantage in comparison with a helical wind turbine with a vertical axis? The motivation for the development of this technology has been added to the introduction section in the second paragraph. The advantage of such device over classical wind turbines has also been explained here.

The performance of the machine are not shown, the CP-lambda curves are not evaluated. The Cp-Lambda curve is an important graph in understanding the performance of the wind turbine. However, due to the following reasons the Cp-Lambda curve was not included:

1. The study conducted was aimed at understanding the aerodynamics of the turbine and the behaviour of wind around the turbine.

2. The current computing system available lacks the computational power to simulate such heavy simulations.

The simple mechanical balance on ducted wind turbines are impenetrable, and a significant original contribution lacks. 1. As the system has one point of inlet and one point of outlet and is enclosed within a duct, there is no mixing or impact of the free-flowing wind. Hence, the energy balance does not reflect the impact of the same considering that the free-flowing wind is impenetrable into the system.

2. The novelty of the research include:

a. The configuration of the system designed.

b. The newly derived equation.

c. The novel working theory 

The bibliography review on ducted wind turbines misses important and recent works revealing the real potential of this kind of devices. Recent publication has been added as per the suggestion.

The use of an ANN is not motivated. The motivation behind using ANN was that it was a computationally affordable method to generate large amount of data to understand the velocity profile trends.

---

## [Decision Letter · Decision Letter 1]

12 Apr 2024

Mathematical Modeling and Performance Evaluation of Ducted Horizontal-axis Helical Wind Turbines: Insights into Aerodynamics and Efficiency

PONE-D-23-38272R1

Dear Dr. Shaikh,

We’re pleased to inform you that your manuscript has been judged scientifically suitable for publication and will be formally accepted for publication once it meets all outstanding technical requirements.

Kind regards,

Ashfaque Ahmed Chowdhury, Ph.D., FHEA, FIEB

Academic Editor

PLOS ONE

Reviewers' comments:

Reviewer's Responses to Questions

**Comments to the Author**

1. If the authors have adequately addressed your comments raised in a previous round of review and you feel that this manuscript is now acceptable for publication, you may indicate that here to bypass the “Comments to the Author” section, enter your conflict of interest statement in the “Confidential to Editor” section, and submit your "Accept" recommendation.

Reviewer #1: All comments have been addressed

Reviewer #2: All comments have been addressed

2. Is the manuscript technically sound, and do the data support the conclusions?

Reviewer #1: Yes

Reviewer #2: Yes

3. Has the statistical analysis been performed appropriately and rigorously? 

Reviewer #1: Yes

Reviewer #2: N/A

4. Have the authors made all data underlying the findings in their manuscript fully available?

Reviewer #1: Yes

Reviewer #2: Yes

5. Is the manuscript presented in an intelligible fashion and written in standard English?

Reviewer #1: Yes

Reviewer #2: Yes

6. Review Comments to the Author

Reviewer #1: All my comments were addressed and no more comments are required to be addressed. The manuscript is ready for publication.

Reviewer #2: (No Response)

7. PLOS authors have the option to publish the peer review history of their article (what does this mean?). If published, this will include your full peer review and any attached files.

Reviewer #1: No

Reviewer #2: No

---

## [Editor Report · Acceptance letter]

29 Apr 2024

PONE-D-23-38272R1 

PLOS ONE

Dear Dr. Shaikh, 

I'm pleased to inform you that your manuscript has been deemed suitable for publication in PLOS ONE. Congratulations! Your manuscript is now being handed over to our production team.

Kind regards, 

on behalf of

Dr. Ashfaque Ahmed Chowdhury 

Academic Editor

PLOS ONE